# Trace Element Deficiency in Systemic Sclerosis—Too Much Effort for Some Traces?

**DOI:** 10.3390/nu16132053

**Published:** 2024-06-27

**Authors:** Daniela Opriș-Belinski, Claudia Oana Cobilinschi, Simona Caraiola, Raluca Ungureanu, Ana-Maria Cotae, Ioana Marina Grințescu, Cristian Cobilinschi, Andrei Cosmin Andrei, Radu Țincu, Răzvan Ene, Liliana Mirea

**Affiliations:** 1Department of Internal Medicine Rheumatology, Carol Davila University of Medicine and Pharmacy, 050474 Bucharest, Romania; daniela.opris@umfcd.ro (D.O.-B.); simona.caraiola@umfcd.ro (S.C.); 2Department of Rheumatology and Internal Medicine, Sfânta Maria Clinical Hospital, 011172 Bucharest, Romania; 3Department of Rheumatology and Internal Medicine, Colentina Clinical Hospital, 020125 Bucharest, Romania; 4Department of Anesthesiology and Intensive Care II, Carol Davila University of Medicine and Pharmacy, 050474 Bucharest, Romania; ralucaung@yahoo.com (R.U.); cotae_ana_maria@yahoo.com (A.-M.C.); ioana.grintescu@umfcd.ro (I.M.G.); cristian.cobilinschi@umfcd.ro (C.C.); andreicosmin1994@gmail.com (A.C.A.); llmirea@yahoo.com (L.M.); 5Department of Anesthesiology and Intensive Care, Clinical Emergency Hospital Bucharest, 014461 Bucharest, Romania; 6Department of Clinical Toxicology, Carol Davila University of Medicine and Pharmacy, 050474 Bucharest, Romania; 7Department of Orthopedics, Carol Davila University of Medicine and Pharmacy, 050474 Bucharest, Romania

**Keywords:** trace elements, systemic sclerosis, scleroderma, iron, selenium, cooper, zinc, malnutrition

## Abstract

Trace elements are essential for several physiological processes. To date, various data have suggested that inadequate levels of trace elements may be involved in the pathogenesis of different chronic diseases, including immune-mediated ones, or may develop during their course. Systemic sclerosis (SSc) is a complex autoimmune multisystemic disease, primarily characterized by microvascular dysregulation, the widespread activation of the immune system and tissue fibrosis. According to the latest reports regarding the pathogenesis of SSc, the main pathophysiological processes—inflammation, vasculopathy and fibrosis—may include various trace element derangements. The present literature review aims to update the available data regarding iron, zinc, copper and selenium status in SSc as well as to underline the possible implications of these trace elements in the complexity of the pathogenic process of the disease. We observe that the status of trace elements in SSc plays a crucial role in numerous pathogenic processes, emphasizing the necessity for proper monitoring and supplementation. The reported data are heterogenous and scarce, and future studies are needed in order to draw clearer conclusions about their complete spectrum.

## 1. Introduction

Systemic sclerosis (SSc) is a complex autoimmune multisystemic disease, primarily characterized by microvascular dysregulation, the widespread activation of the immune system and tissue fibrosis [1,2]. Apart from the skin, which is almost constantly affected in various grades of extension (scleroderma), patients can exhibit visceral involvement such as gastro-intestinal (GI), pulmonary, renal or cardiac. Between 70 and 90% of patients have some degree of digestive involvement, but the most affected part is the lower part of the oesophagus and lower oesophageal sphincter that cause further complications with dysmotility and gastroesophageal reflux. The second most frequent SSc-related GI impairment is gut implication and associated malabsorption [3]. Over the last thirty years, once the understanding of the underlying pathological processes improved, an increase in the survival rate in scleroderma patients was registered [4]. Nevertheless, there is still an increased rate of mortality compared to the general population, especially due to interstitial lung disease, pulmonary hypertension and digestive tract involvement [5]. Regarding the latter, it looks like malnutrition, even if not solely, is the main responsible cause that leads to 8 to 18% excesses of mortality due to gastro-intestinal implications [6].

Malnutrition, defined by the European Society of Clinical Nutrition and Metabolism (ESPEN) as “a state resulting from lack of intake or uptake of nutrition that leads to altered body composition”, has proved to be an independent risk factor for severe outcome in SSc [7,8]. The occurrence of malnutrition in this particular group of patients is considered to be multifactorial (see Figure 1), including anorexia, early satiety, sicca symptoms, oral cavity changes, dysmotility, small intestinal bacterial overgrowth (SIBO) and malabsorption [9,10]. One typical aspect found in scleroderma patients is an evolutive pattern with progressive reduction in the oral aperture (microstomia) and interincisal distance that significantly correlates with poor nutrition capacity [11]. Depression is a frequently identified comorbidity compared to the general population and, as described by Türk et al., it correlates with malnutrition risk [11,12]. Considering the complexity of the disease and different specific features of the disease subtypes, the use of general tools for malnutrition risk screening has proved to be unsatisfactory [13]. As a result, special needs for specific macro- and micronutrients are generally overlooked [14]. 

Trace element abnormalities in patients with SSc may be associated with undernutrition, resulting from decreased food intake or altered intestinal uptake or may be induced by specific pathogenic processes of the disease [7,15]. Although micronutrient deficiencies proved to be more frequent than initially anticipated, conducting a laboratory assessment of micronutrient deficiency is not a routine practice unless it becomes clinically evident [16]. Nonetheless, the obtained laboratory results do not lead to a straightforward decision, considering that micronutrient abnormalities may be influenced by an acute phase response, underlying chronic conditions and related organ dysfunctions [7,17]. Moreover, a clear difference between trace element depletion defined as “concentrations below reference range” and deficiency defined as “concentrations below reference range + clinical or metabolic signs” should be determined [14].

The aim of this review is focused on trace element abnormalities in SSc patients and on the evidence available supporting the rationale of trace element supplementation. 

## 2. Materials and Methods

We performed a systematic literature research based on the Preferred Reporting Items for Systematic Reviews and Meta-Analyses (PRISMA) guidelines. Studies were selected by searching in the PubMed, Scopus and Cochrane databases, including original articles, randomized control trials and observational studies, respectively, in order to identify relevant full texts in English, published clinical trials that included information about the levels of the trace elements iron (Fe), zinc (Zn), selenium (Se) and copper (Cu) and clinical correlations with different organ involvement in patients with SSc (Figure 2). 

We searched using the following keywords: “(systemic sclerosis OR scleroderma AND trace elements)”. To expand our research, we also manually searched the references in relevant studies on SSc and trace elements. This allowed us to identify additional studies for potential inclusion based on their titles. 

Using the PICOS framework (Population/Problem, Intervention, Comparisons, Outcomes and Study design) to formulate research questions and develop search strategies, inclusion and exclusion criteria for the study review were established. The inclusion and exclusion criteria are summarized in Table 1.

The following data were collected from the included studies: the study design, the first author, the year of publication, the sample size, the types of trace elements investigated, the mean values of the investigated trace elements, and trends of the trace elements in the study group.

## 3. Results

We identified 26 articles written in English (8 in PubMed, 18 in Scopus and, respectively, 0 in Cochrane database); furthermore, two researchers screened the titles and abstracts in order to identify relevant articles, and if any disagreements occurred in the selection process, they were settled by a third reviewer. A total of 12 articles were included Figure 2. The reported data of the include articles is summarized in Table 2.

## 4. Discussions

### 4.1. Absolute and/or Functional Iron Deficiency?

Given that iron (Fe) deficiency is considered the most frequent nutritional deficiency worldwide, SSc patients are likewise affected [14,18]. Considering the complex pathogenesis of SSc, patients may develop, over the course of the disease progression, both absolute and functional iron deficiency. 

#### 4.1.1. Iron Status in SSc Population

Reports regarding iron deficiency anemia have been published since the late 1960s, when Westerman et al. indicated that 50% of the investigated cases of SSc with anemia had iron depletion [19].

SSc patients may experience an absolute iron deficiency resulting from a negative iron balance caused by frequent gastrointestinal blood loss as well as due to extended malabsorption or hemolysis [9,20,21]. It has been reported that iron deficiency is more prevalent in SSc patients with pulmonary hypertension (46.1% vs. 16.4%) and is associated with decreased exercise capacity and worse survival rates [22].

Iron deficiency anemia in patients with SSc is usually related to gastrointestinal microhemorrhages or the presence of gastric antral vascular ectasia (GAVE) [23]. A study investigating hematological abnormalities in 180 SSc patients reported that one-third of patients registered iron deficiency anemia, with the main reported cause being gastrointestinal bleeding [23].

Telangiectatic gastrointestinal bleeding is considered one of the main causes of iron store depletion in SSc patients [24,25]. Although the European League Against Rheumatism Scleroderma Trials and Research (EUSTAR) network study conducted by Hughes et al. reported that telangiectasias may be encountered in almost 68% of SSc patients, at the moment, there are no available data regarding the prevalence of telangiectasias affecting the gastrointestinal tract [26].

Another EUSTAR network study, which included the largest SSc cohort, reported a prevalence of symptomatic GAVE of only 1% (*n* = 49); however, not all included patients were systematically assessed using an endoscopy [27]. In a prospective study (*n* = 103) which included SSc patients considered for the Scleroderma: Cyclophosphamide Or Transplant (SCOT) trial, who underwent a screening process with an upper endoscopy, indicated a GAVE prevalence of 22.3% [28]. An analysis of the largest Australian cohort of SSc patients (*n* = 2039) also revealed a higher prevalence of GAVE, reaching 10.6% [29]. More than a half of patients (53.8%) diagnosed with GAVE had already experienced a drop in hemoglobin to 10 g before the moment of the endoscopy [29]. As a result, GAVE is considered one of the leading causes for iron deficiency anemia in SSc patients. According to a prospective comparative study conducted by El-Hawary, who aimed to evaluate different types of therapies for GAVE, iron levels as well as iron metabolism markers were significantly diminished before the intervention [30]. Although a progressive improvement in iron status was reported after cyclophosphamide administration and argon plasma coagulation, no data regarding iron supplementation were reported [30]. 

Considering the central pathogenic processes of the disease, it is suggested that every segment of the gastrointestinal tract may be affected by collagenous fibrosis, causing decreased intestinal permeability and malabsorption [2,31]. Decreased iron intestinal absorption in SSc patients may also be induced by small intestinal bacterial overgrowth (SIBO) [32]. Marie et al. reported that patients with SIBO and increased levels of fecal calprotectin may also be associated with decreased levels of ferritin [33]. 

The extensive use of proton pump inhibitors (PPIs) recommended for the treatment of SSc-related GI reflux contributes to impaired iron absorption [34].

Numerous studies have reported that SSc patients may also exhibit decreased vitamin C levels without a clear underlying cause. However, it can be assumed that it is most probably multifactorial, potentially involving factors such as inadequate intake, gastrointestinal involvement or PPI use [35]. Vitamin C deficit in SSc patients is particularly important given that its role extends beyond the iron absorption process, such as ferritin synthesis and degradation, the modulation of cellular iron efflux or transferrin–iron uptake mechanisms [36].

SSc patients with severe gastrointestinal involvement may often require blood transfusions which may be associated with further iron dysregulation, as free iron provided during transfusions could potentially be used to catalyze inflammation [37].

#### 4.1.2. Potential Impact of Iron on SSc Pathogenesis

Functional iron deficiency may also be encountered in SSc patients and may be directly related to the underlying chronic activation of the immune system [20,38]. 

Since interleukin-6 (IL-6) has a key role in SSc pathogenesis, it is accountable for the increased levels of hepcidin [37,39]. Hepcidin regulates absorption and iron circulation in the body and increases intracellular iron levels by promoting the degradation and internalization of ferroportin [40]. Increased hepcidin values were associated with functional iron deficiency and falsely elevated ferritin [22,37,41]. Ruiter et al. reported both increased hepcidin and IL-6 levels in SSc patients with iron deficiency; however, no association between these two was found [22]. Experimental studies revealed a crosstalk between IL-6 and the bone morphogenetic protein BMP/SMAD pathway, and mutations of BMPR2 were associated with pulmonary hypertension and iron deficiency [40]. However, it has already been demonstrated that SSc patients, even those with associated pulmonary hypertension, do not exhibit BMPR-2 mutations [42]. These findings support the hypothesis that hepcidin expression may be induced by a more complex mechanism than only under the action of IL-6.

According to recent preclinical data regarding the pathogenesis of SSc, iron plays a central role in a distinctive process of programmed cell death called “ferroptosis” [43]. It is considered that excessive ferroptosis cell death is characterized by increased intracellular free ferrous iron accumulation, lipid peroxidation and oxidative stress [43]. 

The dietary reference intake (DRI) of iron should be adapted based on the patient’s gender and age. Currently, the DRI is 8 mg/day, but it should be increased in young pre-menopausal women. In line with the latest ESPEN guideline, treatment for iron deficiency should be initiated based on a comprehensive evaluation of iron status including plasma iron, transferrin, transferrin saturation, ferritin, C-reactive proteins, hepcidin, and red blood cell morphology [14].

### 4.2. Zinc Deficiency—A Potential Risk Factor for Disease Progression?

Zinc (Zn) is a key trace element involved in a variety of cellular processes, such as structural, catalytic, extracellular and intracellular signaling, cell proliferation and apoptosis [44]. Although less studied, zinc deficiency has a high incidence, especially among patients with chronic inflammatory diseases, and is associated with compromised immune systems and increased inflammation [45]. 

#### 4.2.1. Zinc Status in SSc Population

Among SSc patients, one observational, single-center, longitudinal cohort study (*n* = 176 patients) and one retrospective cross-sectional study (*n* = 82 patients) indicated reduced Zn levels ranging from 48% to 10.9% [35,46]. Sun et al. reported, in a case–control study, that total zinc serum levels were lower in SSc patients with associated pulmonary hypertension [47]. Conversely, two case–control studies revealed no significant differences between SSc patients and the control group [48,49]. Risk factors of zinc deficiency in SSc patients may include inadequate intake, malnutrition, gastrointestinal involvement and secondary malabsorption [9,13,14,50].

#### 4.2.2. Potential Impact of Zinc on SSc Pathogenesis

As the intricate pathogenesis of SSc includes inflammatory processes that engage T-cells, macrophages and B-lymphocytes, an associated zinc deficiency which has been linked with various immune disarrangements, especially increased inflammatory mediator release, may impact disease progression [51,52]. An experimental study conducted by Wong et al. demonstrated that zinc deficiency induced a macrophage-like phenotype in a human monocytic cell line THP-1 culture, as well as increased cell adherence capacity and cytokine release [51].

The pathogenesis of SSc is predominantly marked by dysfunctional tissue repair following autoimmune aggression and microvascular injury [2,52]. Transforming growth factor beta (TGF-β) is one of the most extensively documented factors involved in the fibrotic tissue of SSc patients [2,52]. TGF-β proved to exhibit various functions, including the stimulation of fibroblast activity, resulting in increased extracellular matrix production [53]. In the early 2000s, Massague et al. reported that the main mediators of TGF-β signaling activity were SMADs (mother against decapentaplegic proteins) [54]. Moreover, the dysregulated SMAD pathway was proved to play a central role in the fibrotic process in SSc patients [52]. While there are no specific data available concerning the role of zinc in the fibrotic process in SSc patients, it should be noted that zinc modulates the activity of the MG53 protein, a factor involved in wound healing and tissue repair through the inhibition of TGF-β/SMAD signaling [55]. Therefore, it can be assumed that zinc deficiency may be linked to disease progression and severity; however, substantial research data are further required.

Although there are many recommendations regarding the dietary reference intake of zinc, the latest ESPEN guideline proposes 8–15 mg/day of zinc. Decisions for deficiency treatment should be based on zinc plasma levels in conjunction with albumin and C-reactive protein levels [14].

### 4.3. Copper Inadequacy—A Key Element of Exacerbated Oxidative Stress?

Copper (Cu) is an essential trace element functioning as an enzymatic cofactor and interacting with several proteins known as cuproproteins [56]. Various types of copper proteins have been described and classified based on their chemical and geometrical properties (cupredoxins vs. oxidoreductases), as well as on their roles in transporting copper or using it as a cofactor [57].

#### 4.3.1. Copper Status SSc Population

Limited evidence regarding copper levels in SSc patients is available. Sun et al. reported no difference between controls and SSc patients with pulmonary arterial hypertension regarding copper levels [47]. Qayoom et al. found increased levels of both copper and ceruloplasmin in a cross-sectional study which included only twelve patients with SSc and morphea [58]. Similar data were also published by Li et al. who reported a decreased level of copper in SSc patients, especially in those with associated pulmonary fibrosis [15].

The dietary reference intake of copper for healthy adults ranges between 1.1 and 2 mg/day. In SSc patients, both copper deficiency and toxicity are possible and detrimental. Therefore, copper plasmatic levels, ceruloplasmin levels and C-reactive protein levels should be determined before any treatment is initiated [14].

#### 4.3.2. Potential Impact of Copper on SSc Pathogenesis

Copper oxidation–reduction properties (Cu^2+^-Cu^+^) promote electron transfer reactions causing oxidative stress and reactive oxygen species; therefore, the strict control of copper homeostasis is mandatory [56,59].

Decreased Cu-Zn superoxide dismutase (SOD3) activity, which is part of the endogenous defense mechanism against superoxide overproduction, has been shown to be responsible for various diseases characterized by extensive fibrosis. Experimental data from Sun et al. indicated that SOD3 deficiency induces and exacerbates liver fibrogenesis through epithelial–mesenchymal transitions [60]. Moreover, Frank et al. demonstrated that Cu-Zn SOD expression may be mediated by the NO activity [61]. Nevertheless, experimental evidence has revealed the potential antifibrotic effects of Cu-Zn SOD supplementation targeting TGF-β1 overexpression [62]. Very few data reported the use of Cu-Zn SOD supplementation as antifibrotic treatment, especially in patients under radiotherapy [62]. 

### 4.4. Selenium Deficiency—A Promising Central Piece in the SSc Pathogenic Puzzle?

Selenium (Se) is an essential trace element that has a crucial role in thyroid hormone production, the immune system, antioxidant defense, fertility and skeletal development [63]. After intestinal absorption, Selenoprotein P (SELENOP) provides selenium to the other organs. Indeed, the diverse functions of selenium are facilitated through the action of diverse selenoproteins such as glutathione peroxidase, iodothyronine deiodinase or thioredoxin reductase [64].

#### 4.4.1. Selenium Status in SSc Population

One observational longitudinal study which included 176 patients at different stages of SSc reported that selenium deficiency affected 15.6% of the participants, with severity increasing in late stages of the disease [46]. In a retrospective cross-sectional study involving 82 SSc patients, it was indicated that 35% of patients had lower selenium levels which were associated with myocardial dysfunction [35]. These results align with previously published clinical and experimental data, according to which selenium deficiency is associated with decreased exercise capacity and an increased risk of heart failure [65]. 

Because of its presence in various selenoproteins, selenium is involved in redox reactions and the antioxidant system. Several experimental studies confirmed that Se deficiency causes increased levels of reactive oxygen species and decreased glutathione peroxidase activity in mice [66,67]. Moreover, it was reported that selenium deficiency may increase gastrointestinal oxidative stress, DNA damage and apoptosis [68]. In a case–control study conducted by Tikly et al., a reduced selenium level along with decreased global antioxidant activity in SSc patients was reported [48]. Sun et al. conducted a more comprehensive assessment of selenium status by measuring total selenium levels, SELENOP, as well as glutathione peroxidase 3. All three biomarkers were significantly low in patients with severe skin involvement as well as in patients that developed pulmonary arterial hypertension [47].

#### 4.4.2. Potential Impact of Selenium on SSc Pathogenesis

The typical vascular changes encountered in SSc patients, affecting both micro- and macro-circulation, are mainly caused by an unbalanced production of vasoconstricting and vasodilating agents at the endothelial level [53]. At the cellular level, it has been demonstrated that nitric oxide (NO) activity and metabolism is profoundly disrupted, such as NO overproduction, concomitant with diminished NO production by endothelial NO synthase (eNOS) [65]. Although data regarding the role of selenium in SSc pathogenesis are currently limited, the available results regarding vascular pathological changes induced by selenium deficiency, such as decreased NO in its reduced form due to reduced levels of glutathione peroxidase, warrant further research consideration [69,70]. 

The dietary reference intake of selenium varies between 20 μg/day and 90 μg/day and supplementation should be considered after measuring selenium plasma levels, GPX3, albumin levels and C-reactive protein levels [14].

## 5. Conclusions

The role of trace elements in the pathogenesis of different chronic diseases have already been documented. However, due to the lack of high-quality trials, clear procedures for their determination and the subsequent recommendation for their supplementation are not yet established.

Further investigation is warranted for all four analyzed trace elements in order to determine the exact role and the necessity for supplementation in SSc patients. Figure 3 summarizes the potential role of all four studied trace elements in the pathogenesis of SSc. Nevertheless, the roles of other trace elements such as chromium, cobalt, manganese or molybdenum should also be investigated, considering that the physiologic role of trace elements is usually convergent and synergistic.

Taking into account that neither the European Alliance of Associations for Rheumatology (EULAR) nor the EUSTAR network provide any adapted guidance for the management of trace element determination and supplementation, we consider that the ESPEN recommendations should be followed until further data are available.

## Figures and Tables

**Figure 1 nutrients-16-02053-f001:**
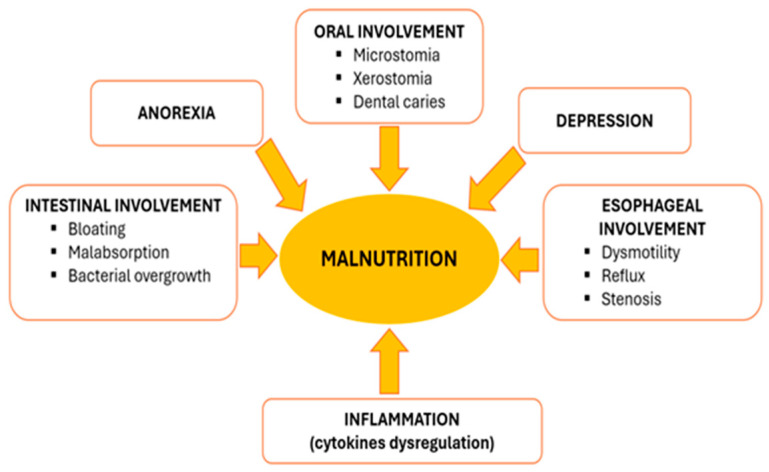
Schematic representation of multifactorial development of malnutrition in SSc patients.

**Figure 2 nutrients-16-02053-f002:**
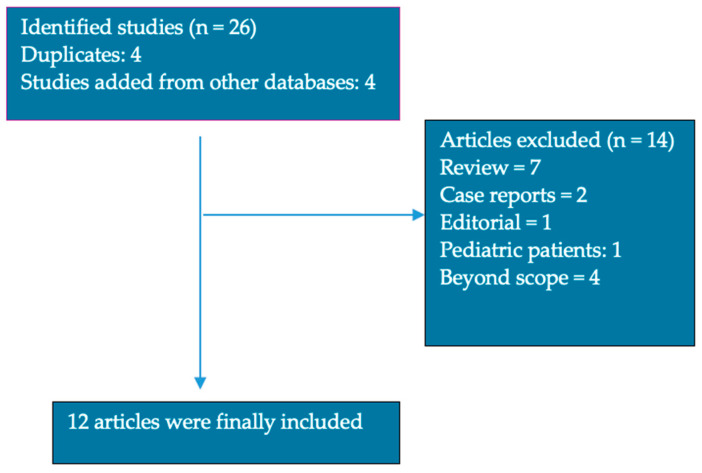
Article selection for review.

**Figure 3 nutrients-16-02053-f003:**
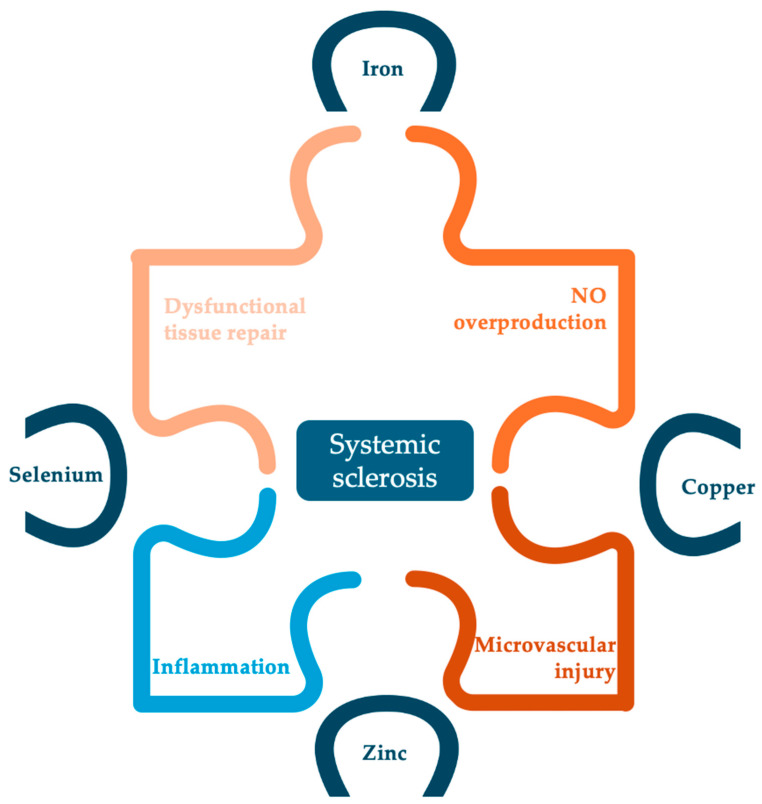
Potential implications of studied trace elements in the pathogenesis of SSc.

**Table 1 nutrients-16-02053-t001:** Inclusion and exclusion criteria.

PICOS Component	Inclusion Criteria	Exclusion Criteria
Population	Patients with SSc aged > 18 years	Patients under 18 years
Intervention	SSc patients with analyzed trace element deficits	SSc patients without any trace element deficiency
Comparisons	Healthy controls without SSc or SSc patients without trace element deficiency	-
Outcomes	Mean level of studied trace elements in SSc patients with deficiencies	-
Study design	Original articles, randomized control trials and observational studies, respectively, such as case–controls or prospective or retrospective cohorts	Editorials, case reports, review articles, published abstracts only

**Table 2 nutrients-16-02053-t002:** Overview of the included studies investigating Fe, Zn, Se and Cu in SSc patients.

Trace Element	Author, Level of Evidence	Patients Sample	Trace Element Change Definition	Average Trace Element in SSc Patients (Mean ± SD)			Trace Element Level Change		
Fe	*Westerman* et al. *(1968)* [12]—retrospective observational study	164 SSc	ID—evaluated based on serum Fe, TIBC and MIS	Serum Fe concentrations ≤46 μg/100 cc			29% SSc—anemia		
				Serum TIBC—<299 mg/100 cc			50% of patients with anemia—Fe store depletion		
	*Frayha* et al. *(1980)* [13]—retrospective observational study	180 SSc	ID anemia was evaluated based on peripheral blood smear, serum Fe, TIBC and MIS	Not available			1/3 SSc—ID		
	*Ruiter* et al. *(2014)* [14]—prospective study	169 SSc	ID—defined by sTfR levels >28.1 nmol/l	SSc-PH			ID—lower exercise capacity in SSC		
		−47 SSc-PH^1^		sTfR = 45.1 ± 14.7 nmol/l			ID prevalence—46.1% in SSc-PH versus 16.4% in SSc-nonPH		
		−122 SSc-nonPH					ID survival rate—HR 0.34, 95% CI 0.14, 0.82, *p* < 0.05 in SSc-PH versus SSc-nonPH patients (HR 0.16, 95% CI 0.02, 1.11, *p*= 0.06)		
				SSc-nonPH					
				sTfR = 34.5 ± 5 nmol/l					
	*El-Hawary* et al. [15] *(2018)*—prospective comparative interventional study	14 SSc with GAVE	ID proven based on Fe, ferritin levels, transferrin saturation and TIBC	Fe = 43.9 ± 5.9 μg/dl			N/A		
				Ferritin = 21.6 ± 3.8 ng/ml					
				Transferrin saturation = 11.7 ± 4.8%					
				TIBC = 511.1 ± 18.8 μg/dl					
	*Xanthouli* et al. [16] *(2023)*—retrospective, cohort study	171 SSc	Ferritin and serum Fe, percentage of hypochromic erythrocytes or red cells (% HRC)	Ferritin = 103.6 ± 128.3 ng/ml			35% SSc—ID		
				Fe = 12.9 ± 6 μmol/L			HRC >2% was associated with lower hemoglobin and Fe levels		
Zn, Se, Cu	*Lundberg* et al. [17] *(1992)*—case–control study	30 SSc	Serum levels measurements for Se, Zn and Cu	Se = 0.86 ± 0.24 μmol/L			Se levels significantly lower in SSc versus control		
				Cu = 20.9 μmol/L			No difference in Zn and Cu levels		
				Zn = 14.1 μmol/L					
	*Sun* et al. [18] *(2020)*—case–control study	66 SSc	*Se status:*	SSc-PH patients:			Se deficiency:		
		30 healthy controls	total serum Se, SELENOP concentrations and GPx3 activity	Serum Se = 91 ± 2 mg/L			-Based on total serum Se: Ssc—14.6% and		
				SELENOP concentrations 3.7 ± 0.8 mg/L			SSc-PH—16%		
			Total serum Zn	GPX3 278 ± 40 U/L			-Based in SELENOP concentrations: SSc—41.5% and SSc-PH—64%		
							-Based on GPX3 activity—SSc—19.5% and SSc-PH—28%.		
			*Cu status:*				Total serum Zn concentration—decreased in SS-PH patients		
			total serum Cu and CP concentrations				Cu concentration—similar in both groups		
							CP levels—elevated in SSc group		
Fe, Zn, Se, Cu	*Tikly* et al. [19] *(2006)*—case–control study	30 SSc	Serum levels measurements for Se, Fe, Zn and Cu	Fe = 22.7 μg/dL			Decreased Se level in SSC versus control group.		
				Zn = 134.1 μg/dL			No difference in Fe, Zn and Cu levels		
				Se = 82.84 μg/L					
				Cu = 148.4 μg/dL					
Fe, Zn, Se	*Marie* et al. [20] *(2014)*—prospective observational cohort study	80 SSc	Ferritin, Zn and Se serum level measurements		*Fructose malabsorbtion*	*No fructose malabsorbtion*	Not reported		
		−32 fructose malabsorbtion		*Ferritin*	82 mmol/L	735 mmol/L			
		−48 no fructose malabsorbtion		*Zn*	11.5 mmol/L	11.9 mmol/L			
				*Se*	0.83 mmol/L	0.86 mmol/L			
Fe, Se	*Dupont* et al. [21] *(2018)*—retrospective cross-sectional study	82 SSc	Ferritin and Se serum level measurements	Ferritin = 94 mg/L			From 55 patients with vitamin C deficiency, 19 (35%) also presented Se deficiency		
				Se = 61 mg/L					

Fe, Cu	*Li* et al. [15] (2020)—observational case–control study	51 SSc	Fe and Cu serum level measurements	Fe = 66.20 ± 22.90 μg/dL			Low concentration of serum Cu was detected in SSc patients in comparison with healthy controls		
		106 healthy controls		Cu = 12.44 ± 2.86 μmol/L					
							Lower concentration of serum Cu was detected in SSc patients with pulmonary fibrosis than without pulmonary fibrosis.		
Se, Zn	*Läubli* et al. [22] *(2020)*—observational, single-center, longitudinal cohort study	250 SSc	Se and Zn level measurement		*Established SSc*	*Very early SSc*		*Established SSc*	*Very early SSc*
		−176 established SSc		*Se*	1.03 μmol/L	1.07 μmol/L	Se deficiency	*15.60%*	*9.10%*
		−74 very early SSc		*Zn*	10.8 μmol/L	10.8 μmol/L	Zn deficiency	*15%*	*10.90%*

SSc—sistemic sclerosis; Fe—iron; Zn—zinc; Se—selenium; Cu—copper; SD—standard deviation; ID—iron deficiency; MIS—marrow iron stores; N/A—not applicable; SSc-PH—SSc-associated pulmonary hypertension; SSc-nonPH—SSc without PH; sTfR—circulating soluble transferrin receptor; GAVE—gastric antral vascular ectasia; TIBC = total iron binding capacity; GPx3—glutathione peroxidase 3; CP—ceruloplasmin.

## Data Availability

Data sharing is not applicable to this article.

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
