# Peer review of "Trace Element Deficiency in Systemic Sclerosis—Too Much Effort for Some Traces?"

_nutrients, 2024, doi:10.3390/nu16132053_

Round 1

Reviewer 1 Report

Comments and Suggestions for Authors

The authors review the existing literature on trace elements in patients with systemic sclerosis. The review is welcome because trace elements have been insufficiently studied in systemic sclerosis and (as the authors point out) may be relevant in different aspects of pathogenesis and (via supplementation) potentially in treatment.

1. The review would benefit from a diagram for each of the 4 trace elements described, indicating the potential role of the trace element in the pathogenesis of systemic sclerosis. This would make the authors’ line of reasoning easier to follow.

2. Some older (but relevant) literature has not been cited.

3. The authors might wish to cite relevant literature from animal models (e.g. selenium deficiency in chickens).

Comments on the Quality of English Language

The authors should check for grammatical and spelling errors. One example = copper (not ‘cooper’).  

Author Response

First, we have to thank you for your time and excellent review and comments that helped us to improve our work. We have incorporated the necessary changes in the revised manuscript point by point based on your comments. We have highlighted the changes in the original manuscript by using blue-coloured text. Also, please find below our responses.

  1. The review would benefit from a diagram for each of the 4 trace elements described, indicating the potential role of the trace element in the pathogenesis of systemic sclerosis. This would make the authors’ line of reasoning easier to follow.

Thank you very much for your suggestion. In order to better organize the extracted information, authors added both a table (Table 2) and figure (Figure 3) that synthetize the reported information. 

  1. Some older (but relevant) literature has not been cited.

Authors have revised available literature and added new reference points with relevance to the presented topic – see reference no 14

  1. The authors might wish to cite relevant literature from animal models (e.g. selenium deficiency in chickens).

Thank you for your suggestion. The manuscript focused more on identifying data on trace elements in SSc, which is already scarce and not homogeneously presented in existing literature. Authors tried to extract essential data in order to formulate ideas on the implication of trace element derangements in SSc with a complex human pathophysiology. Since this is a feasible idea, we will include data reported on animals in our future projects.

Also, please note that as compared to our first version we expanded the part about:

  • systemic sclerosis and its relation with malnutrition and consequently trace elements deficiency, two figures were added (figure 1 and figure 3)
  • materials and methods – inclusion/exclusion criteria (table 1), article selection for review (figure 2)
  • results part was enriched with table number 2 that summarises the available data

Reviewer 2 Report

Comments and Suggestions for Authors

The authors write a review examining evidence of trace elements impact on systemic sclerosis.

This is an interesting topic of relevance.  The authors present some interesting points but the review lacks a degree of organization and comprehensiveness that would be expected.

The authors list keywords used in the review but dot not provide a list databases, number of papers reviewed, and other criteria for inclusion.

The study needs improved organization. I recommend dividing the study into sections based on the major trace elements examined. The study needs to define standardly accepted ranges for each of the trace elements in a clear manner with supporting documentation.  Then, present the quantitative evidence for changes in trace elements in the disease and comparator populations. Finally, hypotheses as to why there are changes in trace elements and their potential impact on the disease should be summarized.

The study lacks clear quantitative results.  There are some intertwined somewhat randomly in amalgamated sentences that summarize individual studies.  However, I would recommend a table organized by trace element that lists number of studies examining it, rough effect size or stated quantitative metric, and a citation for the finding.  Another column might be whether the trace element was increased, decreased, or the same as the general non-diseased comparator population.  Moreover, if there is enough data for the authors to aggregate into aggregate effect sizes, that would be even better. But at least a summary table of quantitative results is warranted.

A figure illustrating the overall concepts addressed in the review and potential relevant mechanisms would engage the readers.  This figure should appear early in the work.

Finally, the tone needs edited in place, particularly the abstract.  It presently reads like there are clear-cut statistical conclusions being drawn by the study, which is presently not done in this review. For example, this sentence, "We have demonstrated that despite the insufficient data dedicated to the trace elements  status in systemic sclerosis they play a crucial role in numerous pathogenic processes emphasizing  the necessity for proper monitoring and supplementation."  The authors did not demonstrate but rather summarized.  I would replace this sentence with a more clear-cut holistic summary findings if possible - which elements are increased, decreased, etc.

MINOR

The authors need to better use organizational headings. 

Additionally, the English formatting needs addressed. Some places there are 1-sentence paragraphs and others there is a mix of sentences that seem almost unrelated that are merged into a paragraph. Each paragraph should have a least 3 sentences with a clearly defined topic and relatedness.

Comments on the Quality of English Language

moderate edits needed

Author Response

First of all we have to thank you for your time and excellent review and comments that helped us to improve our work. We have incorporated the necessary changes in the revised manuscript point by point based on your comments. We have highlighted the changes in the original manuscript by using blue-coloured text. Also, please find below our responses.

The authors write a review examining evidence of trace elements impact on systemic sclerosis. This is an interesting topic of relevance. Thank you very much for your words of appreciation.

The authors present some interesting points but the review lacks a degree of organization and comprehensiveness that would be expected.

The authors list keywords used in the review but dot not provide a list databases, number of papers reviewed, and other criteria for inclusion. based on the major trace elements examined. 

Thank you very much for your suggestion, we have entirely reformulated the material and methods section, including the number or papers, selection process including an illustrative figure (Figure 2), so that the data is more accessible and better organized.

The study needs to define standardly accepted ranges for each of the trace elements in a clear manner with supporting documentation. Then, present the quantitative evidence for changes in trace elements in the disease and comparator populations. 

Thank you very much for your comment, we created Table 2 and reported the suggested data. Moreover, data on recommended supplementation for each trace element was added in the corresponding paragraph.

Finally, hypotheses as to why there are changes in trace elements and their potential impact on the disease should be summarized.

Additional data was formulated in order to address why trace elements play a role in the pathogenesis of SSc. Since these are only hypotheses, the mechanism of action was initially mentioned followed by a probable explanation of their implication in SSc. In order to make data more impactful, a summarizing figure was added (Figure 3). 

The study lacks clear quantitative results. There are some intertwined somewhat randomly in amalgamated sentences that summarize individual studies. However, I would recommend a table organized by trace element that lists number of studies examining it, rough effect size or stated quantitative metric, and a citation for the finding. Another column might be whether the trace element was increased, decreased, or the same as the general non-diseased comparator population. Moreover, if there is enough data for the authors to aggregate into aggregate effect sizes, that would be even better. But at least a summary table of quantitative results is warranted.

The indication was essential in the manuscript structure. We created Table 2 that resolves the matter and addresses the points mentioned by the reviewer.

A figure illustrating the overall concepts addressed in the review and potential relevant mechanisms would engage the readers. This figure should appear early in the work.

Figure 3 was added in the manuscript. It briefly summarizes the potential implications of studied trace elements in the pathogenesis of SSc

Finally, the tone needs edited in place, particularly the abstract. It presently reads like there are clear-cut statistical conclusions being drawn by the study, which is presently not done in this review. For example, this sentence, "We have demonstrated that despite the insufficient data dedicated to the trace elements status in systemic sclerosis they play a crucial role in numerous pathogenic processes emphasizing the necessity for proper monitoring and supplementation." The authors did not demonstrate but rather summarized. I would replace this sentence with a more clear-cut holistic summary findings if possible - which elements are increased, decreased, etc.

Thank you for your suggestion, we edited the abstract, rephrased the ideas and reviewed the whole manuscript for similar editing changes.

Round 2

Reviewer 2 Report

Comments and Suggestions for Authors

The authors have made substantial changes that have addressed the previous critiques. These changes have greatly enhanced the overall value and readability to the intended scientific field.